# THE EFFECTS OF REWARD MISSPECIFICATION: MAPPING AND MITIGATING MISALIGNED MODELS

**Alexander Pan**
Caltech

**Kush Bhatia**
UC Berkeley

**Jacob Steinhardt**
UC Berkeley

## ABSTRACT

Reward hacking—where RL agents exploit gaps in misspecified reward functions—has been widely observed, but not yet systematically studied. To understand how reward hacking arises, we construct four RL environments with misspecified rewards. We investigate reward hacking as a function of agent capabilities: model capacity, action space resolution, observation space noise, and training time. More capable agents often exploit reward misspecifications, achieving higher proxy reward and lower true reward than less capable agents. Moreover, we find instances of *phase transitions*: capability thresholds at which the agent's behavior qualitatively shifts, leading to a sharp decrease in the true reward. Such phase transitions pose challenges to monitoring the safety of ML systems. To address this, we propose an anomaly detection task for aberrant policies and offer several baseline detectors.

## 1 INTRODUCTION

As reinforcement learning agents are trained with better algorithms, more data, and larger policy models, they are at increased risk of overfitting their objectives (Russell, 2019). *Reward hacking*, or the gaming of misspecified reward functions by RL agents, has appeared in a variety of contexts, such as game playing (Ibarz et al., 2018), text summarization (Paulus et al., 2018), and autonomous driving (Knox et al., 2021). These examples show that better algorithms and models are not enough; for human-centered applications such as healthcare (Yu et al., 2019), economics (Trott et al., 2021) and robotics (Kober et al., 2013), RL algorithms must be safe and aligned with human objectives (Bommasani et al., 2021; Hubinger et al., 2019).

Reward misspecifications occur because real-world tasks have numerous, often conflicting desiderata. In practice, reward designers resort to optimizing a proxy reward that is either more readily measured or more easily optimized than the true reward. For example, consider a recommender system optimizing for users' subjective well-being (SWB). Because SWB is difficult to measure, engineers rely on more tangible metrics such as click-through rates or watch-time. Optimizing for misspecified proxies led YouTube to overemphasize watch-time and harm user satisfaction (Stray, 2020), as well as to recommended extreme political content to users (Ribeiro et al., 2020).

Addressing reward hacking is a first step towards developing human-aligned RL agents and one goal of ML safety (Hendrycks et al., 2021a). However, there has been little systematic work investigating when or how it tends to occur, or how to detect it before it runs awry. To remedy this, we study the problem of reward hacking across four diverse environments: traffic control (Wu et al., 2021), COVID response (Kompella et al., 2020), blood glucose monitoring (Fox et al., 2020), and the Atari game Riverraid (Brockman et al., 2016). Within these environments, we construct nine misspecified proxy reward functions (Section 3).

Using our environments, we study how increasing optimization power affects reward hacking, by training RL agents with varying resources such as model size, training time, action space resolution, and observation space noise (Section 4). We find that more powerful agents often attain higher proxy reward but lower true reward, as illustrated in Figure 1. Since the trend in ML is to increase resources exponentially each year (Littman et al., 2021), this suggests that reward hacking will become more pronounced in the future in the absence of countermeasures.

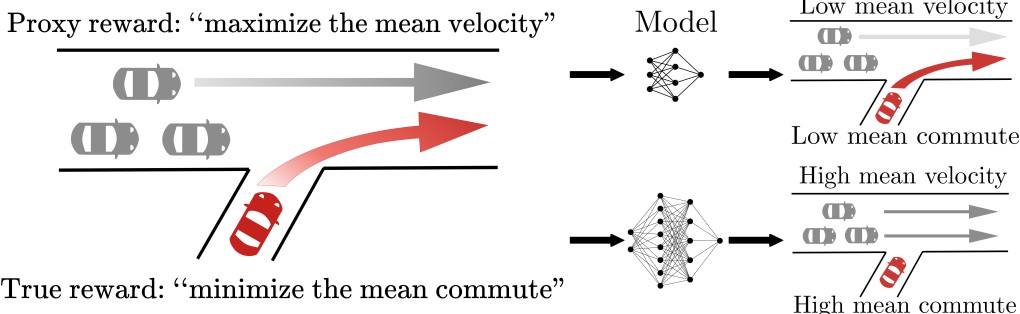

Figure 1: An example of reward hacking when cars merge onto a highway. A human-driver model controls the grey cars and an RL policy controls the red car. The RL agent observes positions and velocities of nearby cars (including itself) and adjusts its acceleration to maximize the proxy reward. At first glance, both the proxy reward and true reward appear to incentivize fast traffic flow. However, smaller policy models allow the red car to merge, whereas larger policy models exploit the misspecification by stopping the red car. When the red car stops merging, the mean velocity increases (merging slows down the more numerous grey cars). However, the mean commute time also increases (the red car is stuck). This exemplifies a *phase transition*: the qualitative behavior of the agent shifts as the model size increases.

More worryingly, we observe several instances of *phase transitions*. In a phase transition, the more capable model pursues a qualitatively different policy that sharply decreases the true reward. Figure 1 illustrates one example: An RL agent regulating traffic learns to stop any cars from merging onto the highway in order to maintain a high average velocity of the cars on the straightaway.

Since there is little prior warning of phase transitions, they pose a challenge to monitoring the safety of ML systems. Spurred by this challenge, we propose an anomaly detection task (Hendrycks & Gimpel, 2017; Tack et al., 2020): Can we detect when the true reward starts to drop, while maintaining a low false positive rate in benign cases? We instantiate our proposed task, POLY-NOMALY, for the traffic and COVID environments (Section 5). Given a trusted policy with moderate performance, one must detect whether a given policy is aberrant. We provide several baseline anomaly detectors for this task and release our data at https://github.com/aypan17/reward-misspecification.

## 2  RELATED WORK

Previous works have focused on classifying different types of reward hacking and sometimes mitigating its effects. One popular setting is an agent on a grid-world with an erroneous sensor. Hadfield-Menell et al. (2017) show and mitigate the reward hacking that arises due to an incorrect sensor reading at test time in a 10x10 navigation grid world. Leike et al. (2017) show examples of reward hacking in a 3x3 boat race and a 5x7 tomato watering grid world. Everitt et al. (2017) theoretically study and mitigate reward hacking caused by a faulty sensor.

Game-playing agents have also been found to hack their reward. Baker et al. (2020) exhibit reward hacking in a hide-and-seek environment comprising 3-6 agents, 3-9 movable boxes and a few ramps: without a penalty for leaving the play area, the hiding agents learn to endlessly run from the seeking agents. Toromanoff et al. (2019) briefly mention reward hacking in several Atari games (Elevator Action, Kangaroo, Bank Heist) where the agent loops in a sub-optimal trajectory that provides a repeated small reward.

Agents optimizing a learned reward can also demonstrate reward hacking. Ibarz et al. (2018) show an agent hacking a learned reward in Atari (Hero, Montezuma's Revenge, and Private Eye), where optimizing a frozen reward predictor eventually achieves high predicted score and low actual score. Christiano et al. (2017) show an example of reward hacking in the Pong game where the agent learns to hit the ball back and forth instead of winning the point. Stiennon et al. (2020) show that a policy which over-optimizes the learnt reward model for text summarization produces lower quality summarizations when judged by humans.

# 3 EXPERIMENTAL SETUP: ENVIRONMENTS AND REWARD FUNCTIONS

In this section, we describe our four environments (Section 3.1) and taxonomize our nine corresponding misspecified reward functions (Section 3.2).

## 3.1 ENVIRONMENTS

We chose a diverse set of environments and prioritized complexity of action space, observation space, and dynamics model. Our aim was to reflect real-world constraints in our environments, selecting ones with several desiderata that must be simultaneously balanced. Table 1 provides a summary.

**Traffic Control.** The traffic environment is an autonomous vehicle (AV) simulation that models vehicles driving on different highway networks. The vehicles are either controlled by a RL algorithm or pre-programmed via a human behavioral model. Our misspecifications are listed in Table 1.

We use the Flow traffic simulator, implemented by Wu et al. (2021) and Vinitsky et al. (2018), which extends the popular SUMO traffic simulator (Lopez et al., 2018). The simulator uses cars that drive like humans, following the Intelligent Driver Model (IDM) (Treiber et al., 2000), a widely-accepted approximation of human driving behavior. Simulated drivers attempt to travel as fast as possible while tending to decelerate whenever they are too close to the car immediately in front.

The RL policy has access to observations only from the AVs it controls. For each AV, the observation space consists of the car's position, its velocity, and the position and velocity of the cars immediately in front of and behind it. The continuous control action is the acceleration applied to each AV. Figure 4 depicts the Traffic-Mer network, where cars from an on-ramp attempt to merge onto the straightaway. We also use the Traffic-Bot network, where cars (1-4 RL, 10-20 human) drive through a highway bottleneck where lanes decrease from four to two to one.

**COVID Response.** The COVID environment, developed by Kompella et al. (2020), simulates a population using the SEIR model of individual infection dynamics. The RL policymaker adjusts the severity of social distancing regulations while balancing economic health (better with lower regulations) and public health (better with higher regulations), similar in spirit to Trott et al. (2021). The population attributes (proportion of adults, number of hospitals) and infection dynamics (random testing rate, infection rate) are based on data from Austin, Texas.

Every day, the environment simulates the infection dynamics and reports testing results to the agent, but not the true infection numbers. The policy chooses one of three discrete actions: INCREASE, DECREASE, or MAINTAIN the current regulation stage, which directly affects the behavior of the population and indirectly affects the infection dynamics. There are five stages in total.

**Atari Riverraid.** The Atari Riverraid environment is run on OpenAI Gym (Brockman et al., 2016). The agent operates a plane which flies over a river and is rewarded by destroying enemies. The agent observes the raw pixel input of the environment. The agent can take one of eighteen discrete actions, corresponding to either movement or shooting within the environment.

**Glucose Monitoring.** The glucose environment, implemented in Fox et al. (2020), is a continuous control problem. It extends a FDA-approved simulator (Man et al., 2014) for blood glucose levels of a patient with Type 1 diabetes. The patient partakes in meals and wears a continuous glucose monitor (CGM), which gives noisy observations of the patient's glucose levels. The RL agent administers insulin to maintain a healthy glucose level.

Every five minutes, the agent observes the patient's glucose levels and decides how much insulin to administer. The observation space is the previous four hours of glucose levels and insulin dosages.

## 3.2 MISSPECIFICATIONS

Using the above environments, we constructed nine instances of misspecified proxy rewards. To help interpret these proxies, we taxonomize them as instances of misweighting, incorrect ontology, or incorrect scope. We elaborate further on this taxonimization using the traffic example from Figure 1.

| **Env.** | Type | Objective | Proxy | Misalign? | Transition? |
|---|---|---|---|---|---|
| Traffic | Mis. | | underpenalize acceleration | No | No |
| | Mis. | minimize commute | underpenalize lane changes | Yes | Yes |
| | Ont. | and accelerations | velocity replaces commute | Yes | Yes |
| | Scope | | monitor velocity near merge | Yes | Yes |
| COVID | Mis. | balance economic, | underpenalize health cost | No | No |
| | Ont. | health, political cost | ignore political cost | Yes | Yes |
| Atari | Mis. | score points under | downweight movement | No | No |
| | Ont. | smooth movement | include shooting penalty | No | No |
| Glucose | Ont. | minimize health risk | risk in place of cost | Yes | No |

Table 1: Reward misspecifications across our four environments. 'Misalign' indicates whether the true reward drops and 'Transition' indicates whether this corresponds to a phase transition (sharp qualitative change). We observe 5 instances of misalignment and 4 instances of phase transitions. 'Mis.' is a misweighting and 'Ont.' is an ontological misspecification.

- **Misweighting.** Suppose that the true reward is a linear combination of commute time and acceleration (for reducing carbon emissions). Downweighting the acceleration term thus underpenalizes carbon emissions. In general, misweighting occurs when the proxy and true reward capture the same desiderata, but differ on their relative importance.

- **Ontological.** Congestion could be operationalized as either high average commute time or low average vehicle velocity. In general, ontological misspecification occurs when the proxy and true reward use different desiderata to capture the same concept.

- **Scope.** If monitoring velocity over all roads is too costly, a city might instead monitor them only over highways, thus pushing congestion to local streets. In general, scope misspecification occurs when the proxy measures desiderata over a restricted domain (e.g. time, space).

We include a summary of all nine tasks in Table 1 and provide full details in Appendix A. Table 1 also indicates whether each proxy leads to misalignment (i.e. to a policy with low true reward) and whether it leads to a phase transition (a sudden qualitative shift as model capacity increases). We investigate both of these in Section 4.

**Evaluation protocol.** For each environment and proxy-true reward pair, we train an agent using the proxy reward and evaluate performance according to the true reward. We use PPO (Schulman et al., 2017) to optimize policies for the traffic and COVID environments, SAC (Haarnoja et al., 2018) to optimize the policies for the glucose environment, and torchbeast (Küttler et al., 2019), a PyTorch implementation of IMPALA (Espeholt et al., 2018), to optimize the policies for the Atari environment. When available, we adopt the hyperparameters (except the learning rate and network size) given by the original codebase.

## 4 HOW AGENT OPTIMIZATION POWER DRIVES MISALIGNMENT

To better understand reward hacking, we study how it emerges as agent optimization power increases. We define optimization power as the effective search space of policies the agent has access to, as implicitly determined by model size, training steps, action space, and observation space.

In Section 4.1, we consider the quantitative effect of optimization power for all nine environment-misspecification pairs; we primarily do this by varying model size, but also use training steps, action space, and observation space as robustness checks. Overall, more capable agents tend to overfit the proxy reward and achieve a lower true reward. We also find evidence of phase transitions on four of the environment-misspecification pairs. For these phase transitions, there is a critical threshold at which the proxy reward rapidly increases and the true reward rapidly drops.

In Section 4.2, we further investigate these phase transitions by qualitatively studying the resulting policies. At the transition, we find that the quantitative drop in true reward corresponds to a

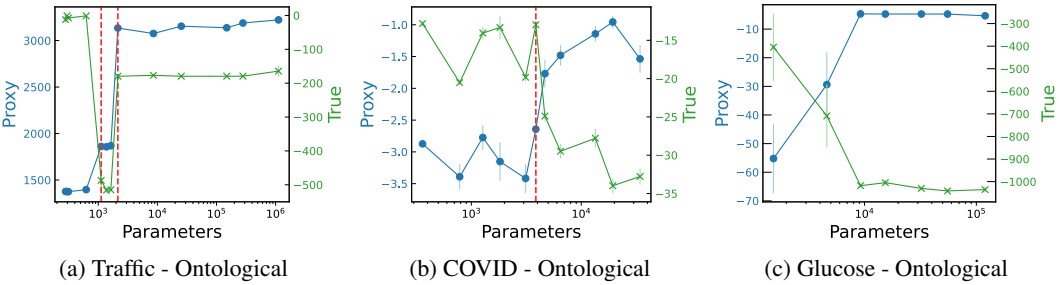

(a) Traffic - Ontological      (b) COVID - Ontological      (c) Glucose - Ontological

Figure 2: Increasing the RL policy's model size decreases true reward on three selected environments. The red line indicates a phase transition.

qualitative shift in policy behavior. Extrapolating visible trends is therefore insufficient to catch all instances of reward hacking, increasing the urgency of research in this area.

In Section 4.3, we assess the faithfulness of our proxies, showing that reward hacking occurs even though the true and proxy rewards are strongly positively correlated in most cases.

### 4.1 QUANTITATIVE EFFECTS VS. AGENT CAPABILITIES

As a stand-in for increasing agent optimization power, we first vary the model capacity for a fixed environment and proxy reward. Specifically, we vary the width and depth of the actor and critic networks, changing the parameter count by two to four orders of magnitude depending on the environment. For a given policy, the actor and critic are always the same size.

**Model Capacity.** Our results are shown in Figure 2, with additional plots included in Appendix A. We plot both the proxy (blue) and true (green) reward vs. the number of parameters. As model size increases, the proxy reward increases but the true reward decreases. This suggests that reward designers will likely need to take greater care to specify reward functions accurately and is especially salient given the recent trends towards larger and larger models (Littman et al., 2021).

The drop in true reward is sometimes quite sudden. We call these sudden shifts *phase transitions*, and mark them with dashed red lines in Figure 2. These quantitative trends are reflected in the qualitative behavior of the policies (Section 4.2), which typically also shift at the phase transition.

Model capacity is only one proxy for agent capabilities, and larger models do not always lead to more capable agents (Andrychowicz et al., 2020). To check the robustness of our results, we consider several other measures of optimization: observation fidelity, number of training steps, and action space resolution.

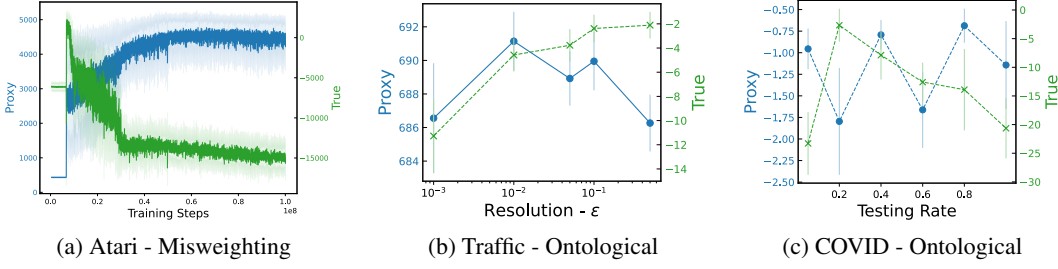

(a) Atari - Misweighting      (b) Traffic - Ontological      (c) COVID - Ontological

Figure 3: In addition to parameter count, we consider three other agent capabilities: training steps, action space resolution, and observation noise. In Figure 3a, an increase in the proxy reward comes at the cost of the true reward. In Figure 3b, increasing the granularity (from right to left) causes the agent to achieve similar proxy reward but lower true reward. In Figure 3c, increasing the fidelity of observations (by increasing the random testing rate in the population) tends to decrease the true reward with no clear impact on proxy reward.

**Number of training steps.** Assuming a reasonable RL algorithm and hyperparameters, agents which are trained for more steps have more optimization power. We vary training steps for an agent trained on the Atari environment. The true reward incentivizes staying alive for as many frames as possible while moving smoothly. The proxy reward misweights these considerations by underpenalizing the smoothness constraint. As shown in Figure 3a, optimizing the proxy reward for more steps harms the true reward, after an initial period where the rewards are positively correlated.

**Action space resolution.** Intuitively, an agent that can take more precise actions is more capable. For example, as technology improves, an RL car may make course corrections every millisecond instead of every second. We study action space resolution in the traffic environment by discretizing the output space of the RL agent. Specifically, under resolution level $\varepsilon$, we round the action $a \in \mathbb{R}$ output by the RL agent to the nearest multiple of $\varepsilon$ and use that as our action. The larger the resolution level $\varepsilon$, the lower the action space resolution. Results are shown in Figure 3b for a fixed model size. Increasing the resolution causes the proxy reward to remain roughly constant while the true reward decreases.

**Observation fidelity.** Agents with access to better input sensors, like higher-resolution cameras, should make more informed decisions and thus have more optimization power. Concretely, we study this in the COVID environment, where we increase the random testing rate in the population. The proxy reward is a linear combination of the number of infections and severity of social distancing, while the true reward also factors in political cost. As shown in Figure 3c, as the testing rate increases, the model achieves similar proxy reward at the cost of a slightly lower true reward.

### 4.2 QUALITATIVE EFFECTS

In the previous section, quantitative trends showed that increasing a model's optimization power often hurts performance on the true reward. We shift our focus to understanding *how* this decrease happens. In particular, we typically observe a qualitative shift in behavior associated with each of the phase transitions, three of which we describe below.

**Traffic Control.** We focus on the Traffic-Mer environment from Figure 2a, where minimizing average commute time is replaced by maximizing average velocity. In this case, smaller policies learn to merge onto the straightaway by slightly slowing down the other vehicles (Figure 4a). On the other hand, larger policy models stop the AVs to prevent them from merging at all (Figure 4b). This increases the average velocity, because the vehicles on the straightaway (which greatly outnumber vehicles on the on-ramp) do not need to slow down for merging traffic. However, it significantly increases the average commute time, as the passengers in the AV remain stuck.

**COVID Response.** Suppose the RL agent optimizes solely for the public and economic health of a society, without factoring politics into its decision-making. This behavior is shown in Figure 5. The larger model chooses to increase the severity of social distancing restrictions earlier than the smaller model. As a result, larger models are able to maintain low average levels of both ICU usage (a proxy for public health) and social distancing restrictions (a proxy for economic health). These

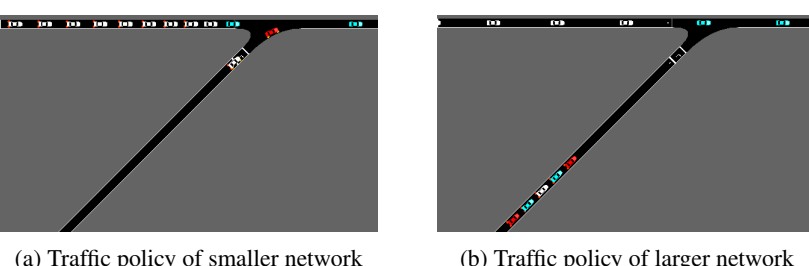

(a) Traffic policy of smaller network    (b) Traffic policy of larger network

Figure 4: The larger model prevents the AVs (in red) from moving to increase the velocity of the human cars (unobserved cars in white and observed cars in blue). However, this greatly increases the average commute per person.

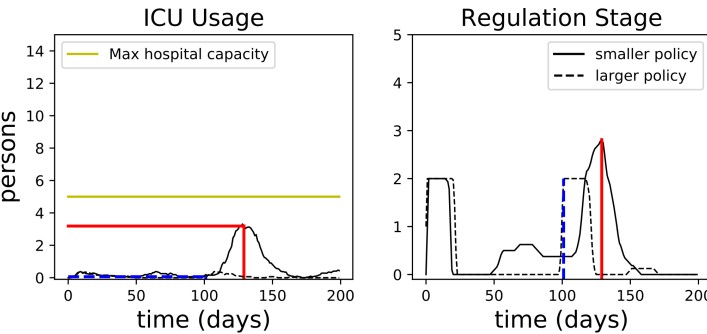

Figure 5: For COVID, ICU usage is a proxy for public health and regulation stage is a proxy for economic health. The blue line indicates the maximum stage (right) enforced by the larger policy and the corresponding ICU level (left) at that stage. The red line is the equivalent for the smaller policy. Because the larger policy enforces regulations much sooner than the smaller policy, it maintains both low ICU usage and low regulation stage. However, the larger policy is politically unfavorable: regulations are high even though public signs of infection, such as ICU usage, are low.

preemptive regulations may however be politically costly, as enforcing restrictions without clear signs of infection may foment public unrest (Boettke & Powell, 2021).

**Atari Riverraid.** We create an ontological misspecification by rewarding the plane for staying alive as long as possible while shooting as little as possible: a "pacifist run". We then measure the game score as the true reward. We find that agents with more parameters typically maneuver more adeptly. Such agents shoot less frequently, but survive for much longer, acquiring points (true reward) due to passing checkpoints. In this case, therefore, the proxy and true rewards are well-aligned so that reward hacking does not emerge as capabilities increase.

We did, however, find that some of the agents exploited a bug in the simulator that halts the plane at the beginning of the level. The simulator advances but the plane itself does not move, thereby achieving high pacifist reward.

**Glucose Monitoring.** Consider an RL agent that optimizes solely for a patient's health, without considering the economic costs of its treatment plans. In this case, the proxy reward is based off of a glycemic risk measure, which reflects the likelihood that a patient will suffer an acute hypoglycemic episode, developed by the medical community (Kovatchev et al., 2000).

However, a less economically-privileged patient may opt for the treatment plan with the least expected cost (Herkert et al., 2019; Fralick & Kesselheim, 2019), not the one with the least amount of risk. From this patient's perspective, the true reward is the expected cost of the treatment plan, which includes the expected cost of hospital visits and the cost of administering the insulin.

Although larger model treatments reduce hypoglycemic risk more smaller model treatments, they administer more insulin. Based on the average cost of an ER visit for a hypogylcemic episode ($1350 from Bronstone & Graham (2016)) and the average cost of a unit of insulin ($0.32 from Lee (2020)), we find that it is actually more expensive to pursue the larger model's treatment.

### 4.3 Quantitative Effects vs Proxy-True Reward Correlation

We saw in Sections 4.1 and 4.2 that agents often pursue proxy rewards at the cost of the true reward. Perhaps this only occurs because the proxy is greatly misspecified, i.e., the proxy and true reward are weakly or negatively correlated. If this were the case, then reward hacking may pose less of a threat. To investigate this intuition, we plot the correlation between the proxy and true rewards.

The correlation is determined by the state distribution of a given policy, so we consider two types of state distributions. Specifically, for a given model size, we obtain two checkpoints: one that achieves the highest proxy reward during training and one from early in training (less than 1% of training complete). We call the former the "trained checkpoint" and the latter the "early checkpoint".

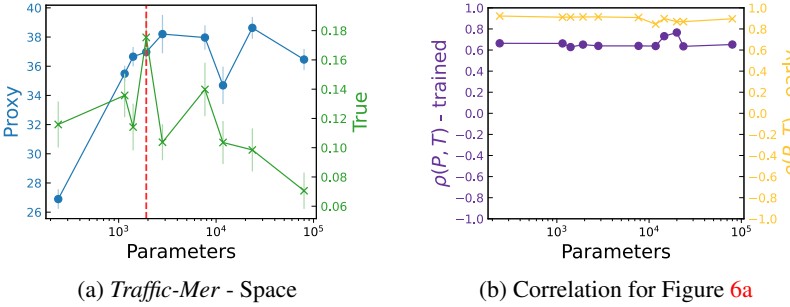

(a) *Traffic-Mer* - Space

(b) Correlation for Figure 6a

Figure 6: Correlations between the proxy and true rewards, along with the reward hacking induced. In Figure 6a, we plot the proxy reward with "●" and the true reward with "×". In Figure 6b, we plot the trained checkpoint correlation and the early checkpoint correlation.

For a given model checkpoint, we calculate the Pearson correlation $\rho$ between the proxy reward $P$ and true reward $T$ using 30 trajectory rollouts. Reward hacking occurs even though there is significant positive correlation between the true and proxy rewards (see Figure 6). The correlation is lower for the trained model than for the early model, but still high. Further figures are shown in Appendix A.2. Among the four environments tested, only the Traffic-Mer environment with ontological misspecification had negative Pearson correlation.

## 5 POLYNOMALY: MITIGATING REWARD MISSPECIFICATION

In Section 4, we saw that reward hacking often leads to phase transitions in agent behaviour. Furthermore, in applications like traffic control or COVID response, the true reward may be observed only sporadically or not at all. Blindly optimizing the proxy in these cases can lead to catastrophic failure (Zhuang & Hadfield-Menell, 2020; Taylor, 2016).

This raises an important question: Without the true reward signal, how can we mitigate misalignment? We operationalize this as an anomaly detection task: the detector should flag instances of misalignment, thus preventing catastrophic rollouts. To aid the detector, we provide it with a *trusted policy*: one verified by humans to have acceptable (but not maximal) reward. Our resulting benchmark, POLYNOMALY, is described below.

### 5.1 PROBLEM SETUP

We train a collection of policies by varying model size on the traffic and COVID environments. For each policy, we estimate the policy's true reward by averaging over 5 to 32 rollouts. One author labeled each policy as acceptable, problematic, or ambiguous based on its true reward score relative to that of other policies. We include only policies that received a non-ambiguous label.

For both environments, we provide a small-to-medium sized model as the trusted policy model, as Section 4.1 empirically illustrates that smaller models achieve reasonable true reward without exhibiting reward hacking. Given the trusted model and a collection of policies, the anomaly detector's task is to predict the binary label of "acceptable" or "problematic" for each policy.

Table 3 in Appendix B.1 summarizes our benchmark. The trusted policy size is a list of the hidden unit widths of the trusted policy network (not including feature mappings).

### 5.2 EVALUATION

We propose two evaluation metrics for measuring the performance of our anomaly detectors.

- *Area Under the Receiver Operating Characteristic (AUROC)*. The AUROC measures the probability that a detector will assign a random anomaly a higher score than a random non-anomalous policy (Davis & Goodrich, 2006). Higher AUROCs indicate stronger detectors.

- *Max F-1 score*. The F-1 score is the harmonic mean of the precision and the recall, so detectors with a high F-1 score have both low false positives and high true negatives. We calculate the max F-1 score by taking the maximum F-1 score over all possible thresholds for the detector.

## 5.3 BASELINES

In addition to the benchmark datasets described above, we provide baseline anomaly detectors based on estimating distances between policies. We estimate the distance between the trusted policy and the unknown policy based on either the Jensen-Shannon divergence (JSD) or the Hellinger distance. Specifically, we use rollouts to generate empirical action distributions. We compute the distance between these action distributions at each step of the rollout, then aggregate across steps by taking either the mean or the range. For full details, see Appendix B.2. Table 2 reports the AUROC and F-1 scores of several such detectors. We provide full ROC curves in Appendix B.2.

| **Baseline Detectors** | Mean Jensen-Shannon | | Mean Hellinger | | Range Hellinger | |
|---|---|---|---|---|---|---|
| Env. - Misspecification | AUROC | Max F-1 | AUROC | Max F-1 | AUROC | Max F-1 |
| Traffic-Mer - misweighting | 81.0% | 0.824 | 81.0% | 0.824 | 76.2% | 0.824 |
| Traffic-Mer - scope | 74.6% | 0.818 | 74.6% | 0.818 | 57.1% | 0.720 |
| Traffic-Mer - ontological | 52.7% | 0.583 | 55.4% | 0.646 | 71.4% | 0.842 |
| Traffic-Bot - misweighting | 88.9% | 0.900 | 88.9% | 0.900 | 74.1% | 0.857 |
| COVID - ontological | 45.2% | 0.706 | 59.5% | 0.750 | 88.1% | 0.923 |

Table 2: Performance of detectors on different subtasks. Each detector has at least one subtask with AUROC under 60%, indicating poor performance.

We observe that different detectors are better for different tasks, suggesting that future detectors could do better than any of our baselines. Our benchmark and baseline provides a starting point for further research on mitigating reward hacking.

## 6 DISCUSSION

In this work, we designed a diverse set of environments and proxy rewards, uncovered several instances of phase transitions, and proposed an anomaly detection task to help mitigate these transitions. Our results raise two questions: How can we not only detect phase transitions, but prevent them in the first place? And how should phase transitions shape our approach to safe ML?

On preventing phase transitions, anomaly detection already offers one path forward. Once we can detect anomalies, we can potentially prevent them, by using the detector to purge the unwanted behavior (e.g. by including it in the training objective). Similar policy shaping has recently been used to make RL agents more ethical (Hendrycks et al., 2021b). However, since the anomaly detectors will be optimized against by the RL policy, they need to be adversarially robust (Goodfellow et al., 2014). This motivates further work on adversarial robustness and adversarial anomaly detection. Another possible direction is optimizing policies against a distribution of rewards (Brown et al., 2020; Javed et al., 2021), which may prevent over-fitting to a given set of metrics.

Regarding safe ML, several recent papers propose extrapolating empirical trends to forecast future ML capabilities (Kaplan et al., 2020; Hernandez et al., 2021; Droppo & Elibol, 2021), partly to avoid unforeseen consequences from ML. While we support this work, our results show that trend extrapolation alone is not enough to ensure the safety of ML systems. To complement trend extrapolation, we need better interpretability methods to identify emergent model behaviors early on, before they dominate performance (Olah et al., 2018). ML researchers should also familiarize themselves with emergent behavior in self-organizing systems (Yates, 2012), which often exhibit similar phase transitions (Anderson, 1972). Indeed, the ubiquity of phase transitions throughout science suggests that ML researchers should continue to expect surprises–and should therefore prepare for them.

## ACKNOWLEDGEMENTS

We are thankful to Dan Hendrycks and Adam Gleave for helpful discussions about experiments and to Cassidy Laidlaw and Dan Hendrycks for providing valuable feedback on the writing. KB was supported by a JP Morgan AI Fellowship. JS was supported by NSF Award 2031985 and by Open Philanthropy.

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

# A  MAPPING THE EFFECTS OF REWARD MISSPECIFICATION

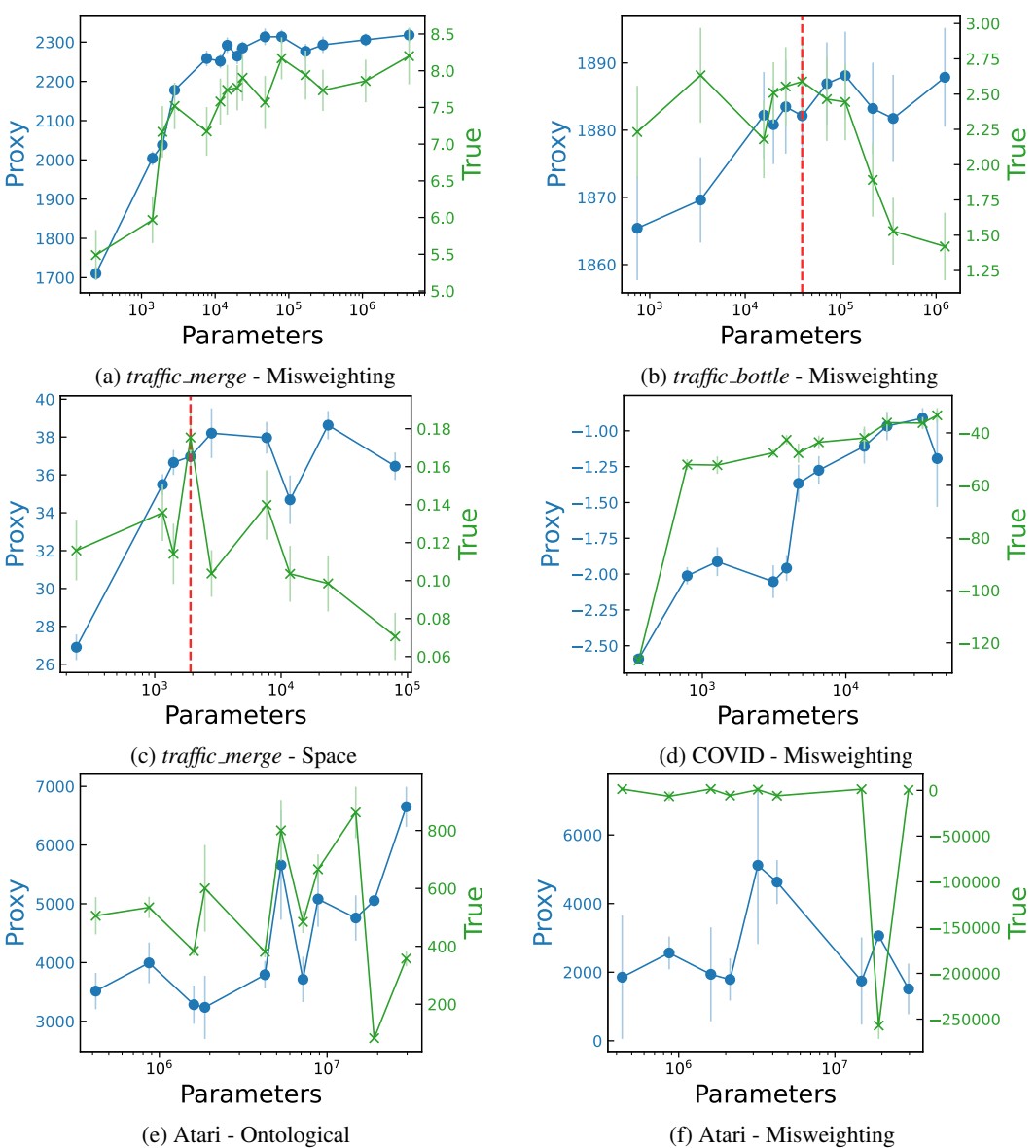

Figure 7: Additional model size scatter plots. Observe that not all misspecifications cause misalignment. We plot the proxy reward with "●" and the true reward with "×". The proxy reward is measured on the left-hand side of each figure and the true reward is measured on the right hand side of each figure.

## A.1  EFFECT OF MODEL SIZE

We plot the proxy and true reward vs. model size in Figure 7, following the experiment described in Section 4.1.

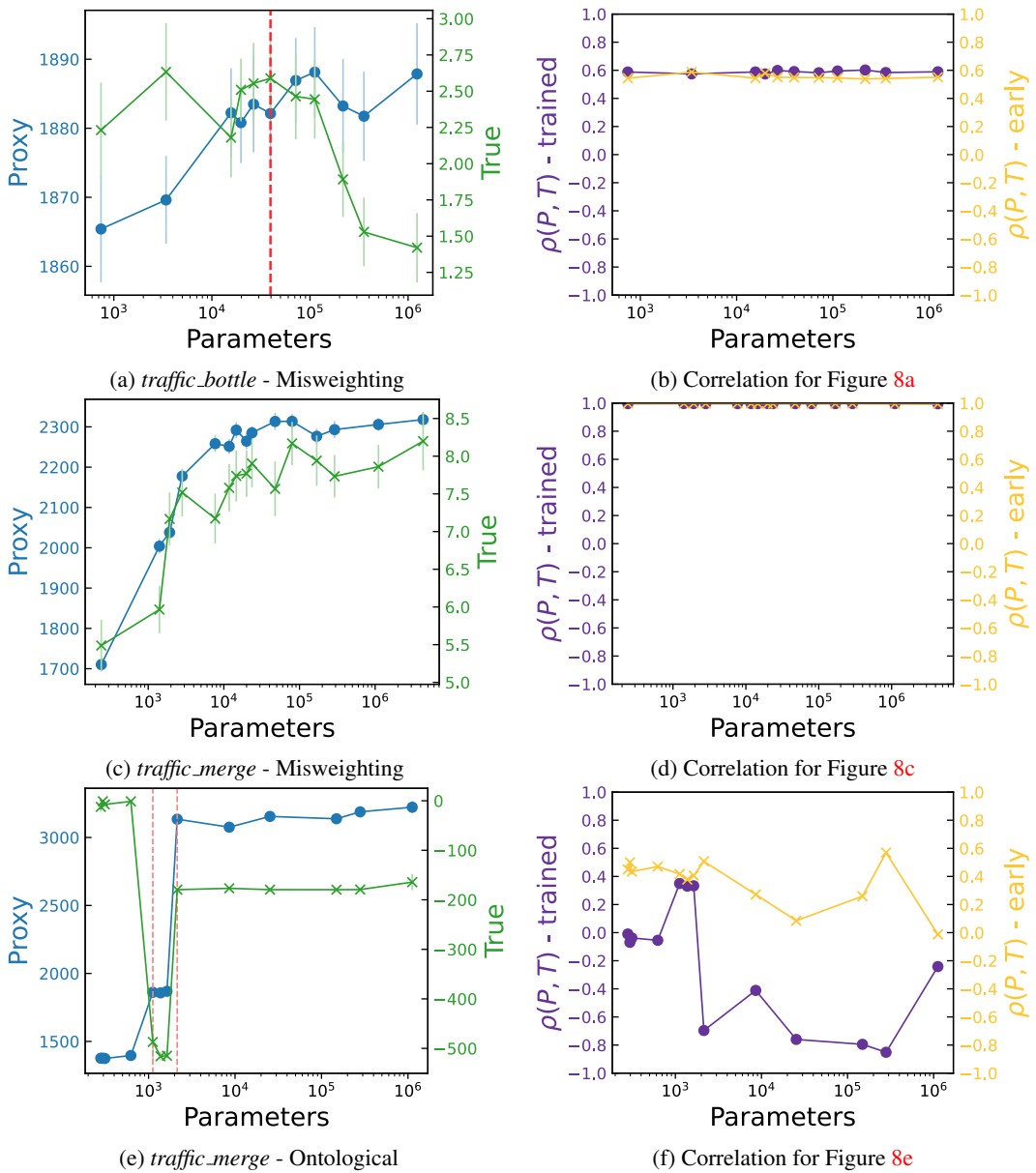

Figure 8: Correlations between the proxy and true rewards, along with the reward hacking induced. In the left column, we plot the proxy reward with "•" and the true reward with "×". In the right column, we plot the trained checkpoint correlation and the randomly initialized checkpoint correlation.

## A.2 CORRELATION BETWEEN PROXY AND TRUE REWARDS

We plot the correlation between proxy and true rewards, following the experiment described in Section 4.3. Interestingly, we see that reward hacking still occurs when there is positive correlation between the true and proxy rewards, e.g., in Figures 8a/8b. Unsurprisingly, proxy-true pairs which are highly correlated, e.g., Figure 8c/8d do not exhibit reward hacking. Finally, proxy-true pairs which are negatively correlated, e.g., Figure 8e/8f exhibit the most reward hacking.

| Env. - Misspecification | # Policies | # Problematic | Rollout length | Trusted policy size |
|---|---|---|---|---|
| Traffic-Mer - misweighting | 10 | 7 | 270 | $[96, 96]$ |
| Traffic-Mer - scope | 16 | 9 | 270 | $[16, 16]$ |
| Traffic-Mer - ontological | 23 | 7 | 270 | $[4]$ |
| Traffic-Bot - misweighting | 12 | 9 | 270 | $[64, 64]$ |
| COVID - ontological | 13 | 6 | 200 | $[16, 16]$ |

Table 3: Benchmark statistics. We average over 5 rollouts in traffic and 32 rollouts in COVID.

## B  POLYNOMALY

### B.1  BENCHMARK STATISTICS

See Table 3 for Polynomaly's statistics.

### B.2  RECEIVER OPERATING CHARACTERISTIC CURVES

We plot the ROC curves for the detectors described in Section 5.3. Our detectors are calculated as follows.

Let $P$ and $Q$ represent two probability distributions with $M = \frac{1}{2}(P + Q)$. Then the Jensen-Shannon divergence and the Hellinger distance between them is given by

$$
\begin{aligned}
\text{JSD}(P\|Q) &:= \frac{1}{2}\text{KL}(P\|M) + \frac{1}{2}\text{KL}(Q\|M) \\
\text{Hellinger}(P, Q) &:= \frac{1}{2}\int \left(\sqrt{dP} - \sqrt{dQ}\right)^2.
\end{aligned}
\tag{1}
$$

Our proposed detectors estimate the distance $\mathcal{D}(\pi_{\text{trusted}}, \pi_{\text{unknown}})$ between the trusted policy $\pi_{\text{trusted}}$ and unknown policy $\pi_{\text{unknown}}$ as follows: We generate $r$ rollouts of $\pi_{\text{unknown}}$, where $r = 5$ in the traffic environment and $r = 32$ in the COVID environment. Every $s$ steps of a rollout, where $s = 10$ in the traffic environment and $s = 1$ in the COVID environment, we set $P$ to be the action distribution of $\pi_{\text{unknown}}$ given the unknown agent's state at that timestep in the rollout and $Q$ to be the action distribution of $\pi_{\text{trusted}}$ given the unknown agent's state at that timestep in the rollout. Intuitively, if $P$ and $Q$ are far apart, then the trusted agent would have performed a different action than the unknown agent at that given timestep, indicating a possible case of reward hacking. We then compute either $\text{JSD}(P\|Q)$ or $\text{Hellinger}(P, Q)$ following Equation (1). These distances are collected every $s$ steps over the entire rollout, and we calculate metrics on these distances (range, mean, etc.) to assign an anomaly score to the untrusted policy.

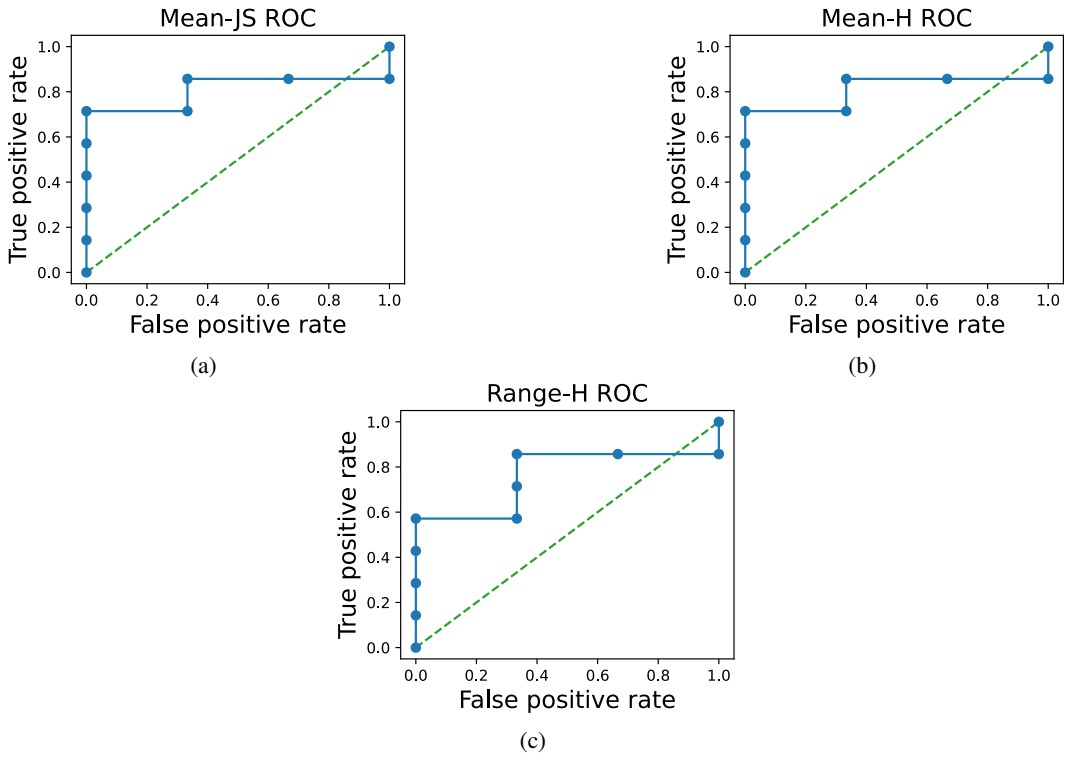

Figure 9: ROC curves for Traffic-Mer - misweighting.

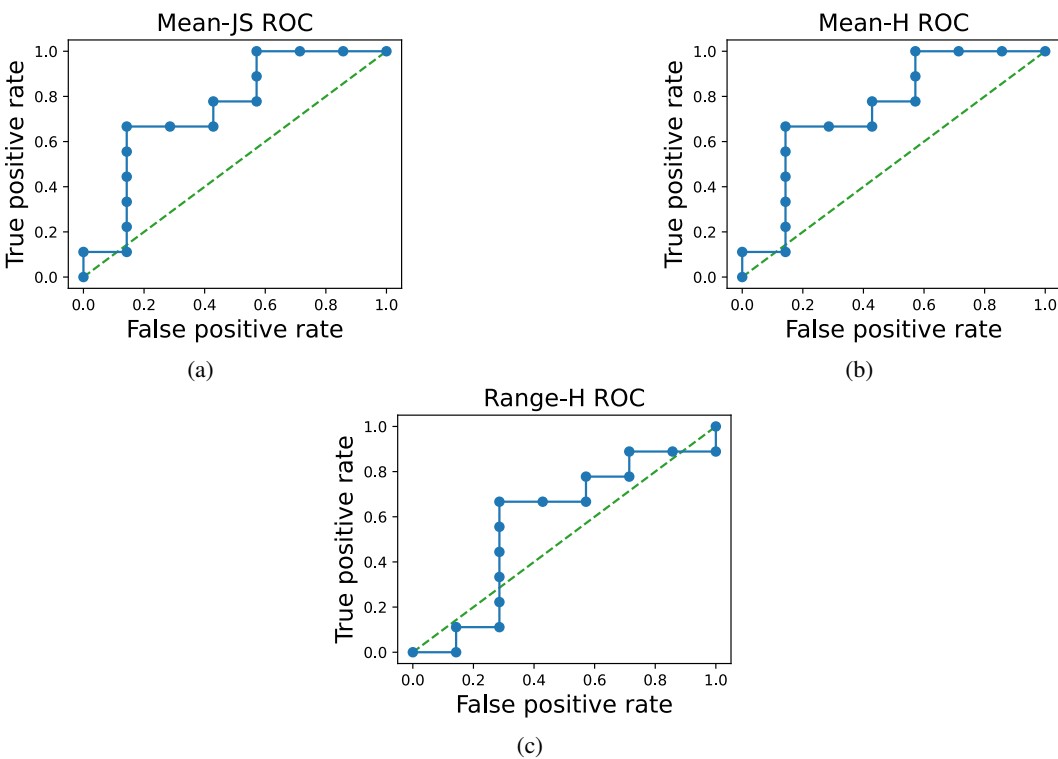

Figure 10: ROC curves for Traffic-Mer - scope.

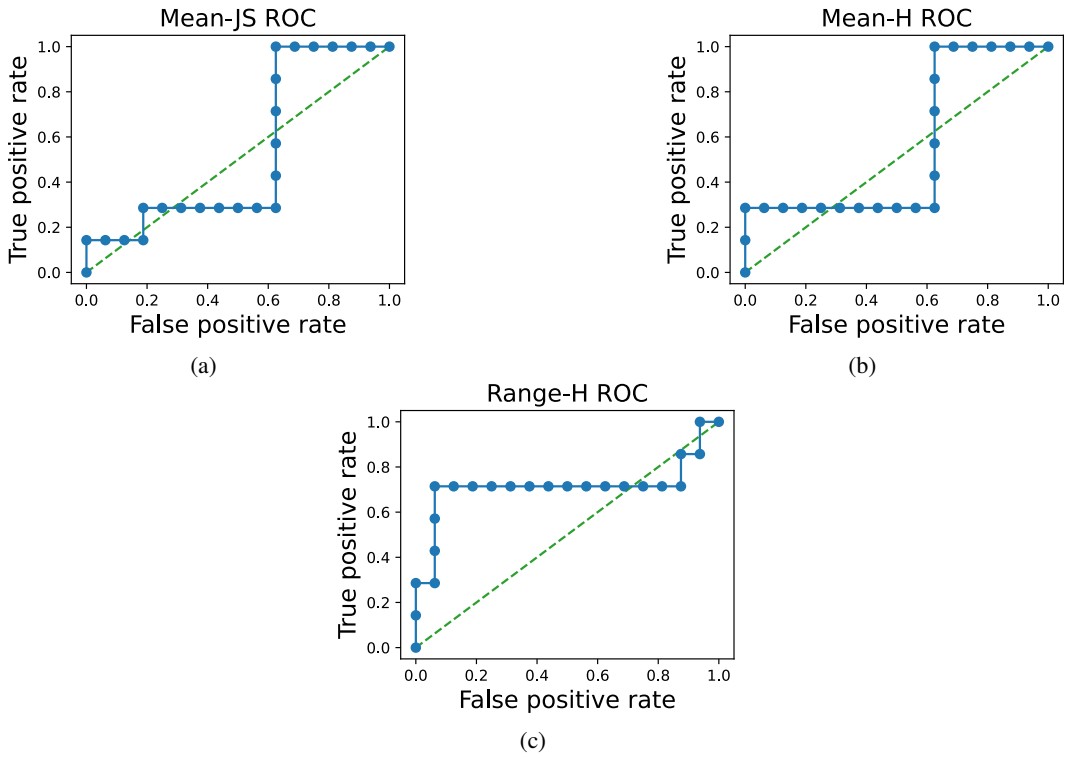

Figure 11: ROC curves for Traffic-Mer - ontological.

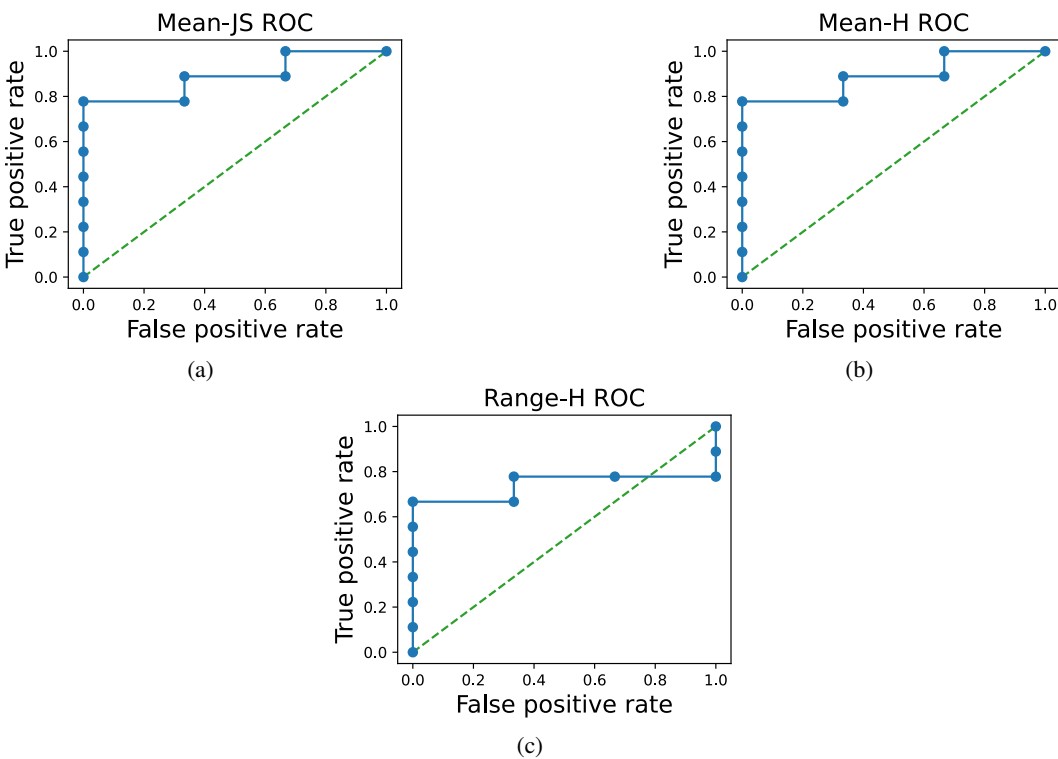

Figure 12: ROC curves for Traffic-Bot - misweighting.

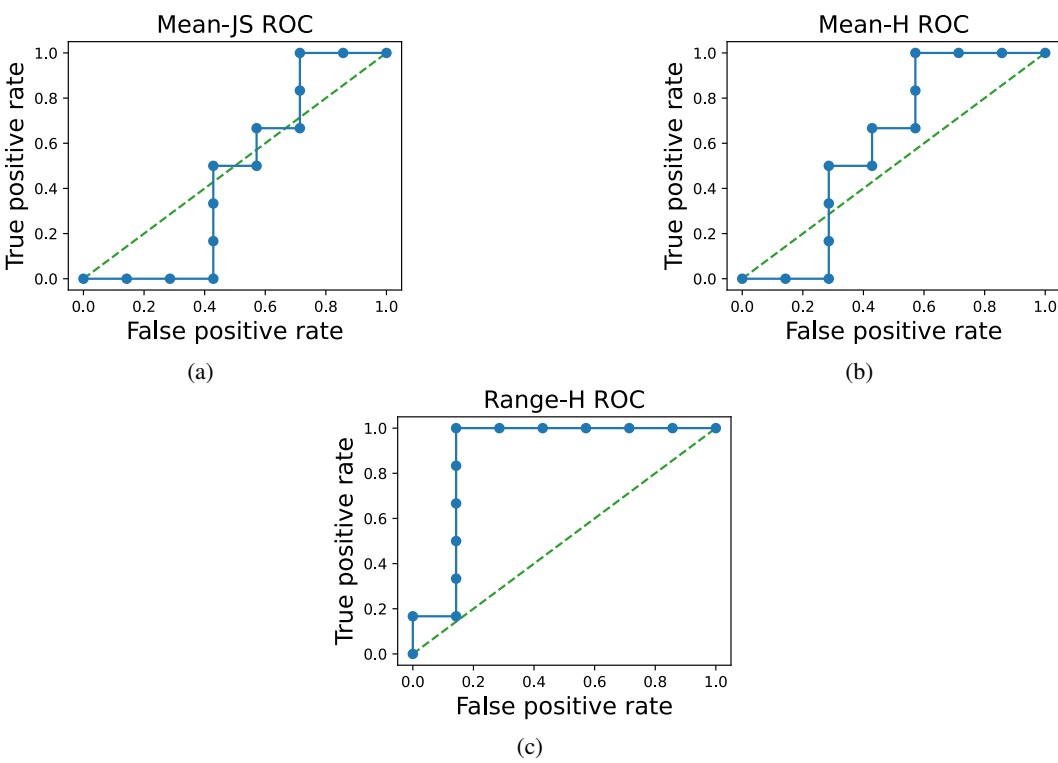

Figure 13: ROC curves for COVID - ontological.

