# OpenReview forum: "The Effects of Reward Misspecification: Mapping and Mitigating Misaligned Models"
_ICLR.cc/2022/Conference — ICLR 2022 Poster_

### Official Review · Reviewer_16uL · 2021-10-29

**Correctness:** 4
**Technical Novelty And Significance:** 3
**Empirical Novelty And Significance:** 4
**Recommendation:** 8
**Confidence:** 4

**Main Review:**

This paper does a great job at investigating an important problem in AI safety in a concrete and rigorous way. There is an informal assumption in AI safety that the reward hacking problem will tend to get worse as the agent becomes more capable. To my knowledge, this paper is the first to demonstrate this effect quantitatively by varying different agent capabilities (such as model capacity) and showing that the true reward decreases as a result. The paper is clearly written and well-motivated.

This paper systematically investigates the reward hacking phenomenon across a variety of environments, agent capabilities and forms of misspecification. The environments used in the paper are diverse and complex, which helps to show that reward hacking occurs in a wide variety of domains (though it would be good to see results on more than one Atari environment). The environments were thoughtfully chosen to include tradeoffs between several desiderata that have to be managed by the agent.

The authors introduce a taxonomy of misspecification (misweighting, ontology and scope) and design a variety of proxy rewards illustrating the different forms of misspecification. The choice of proxy and true rewards for some of the environments seemed a bit arbitrary (e.g. average speed vs mean commute time), so it would be great to see more justification for this.

I found the section on mitigating reward misspecification to be a somewhat weaker point of the paper. I have the impression that the proposed benchmark requires a large amount of human feedback - it would be great to include the human time cost for the anomaly detection task. The anomaly detection task is mostly illustrated for the traffic environment - it would be useful to include more of the other environments as well. I also wonder whether using the distance from the (suboptimal) trusted policy would result in labeling a policy with superhuman performance on the true reward as aberrant (e.g. AlphaGo's Move 37, which is very unlikely for a human player).



**Summary Of The Paper:**

This paper investigates the phenomenon of reward hacking as a function of agent capabilities. They introduce four diverse RL environments with nine misspecified rewards and demonstrate that more capable agents are better at exploiting the misspecification. They find instances of phase transitions where a small increase in agent capability produces a large change in behavior that sharply decreases the true reward.

To mitigate the reward hacking problem, they propose to set up an anomaly detection task, given a trusted model with moderate performance on the true reward, where the anomaly detector's task is to identify whether policies from a different model are satisfactory for the true reward. They provide several baseline anomaly detectors and show how they perform on different tasks.

**Summary Of The Review:**

This paper does a great job at investigating an important problem in AI safety in a concrete and rigorous way, and to my knowledge is the first to do so. I am in favour of accepting this paper.

---

> ### Author Response · Authors · 2021-11-18
> **Response to Reviewer 16uL**
>
> Thank you for your suggestions! We are glad that you appreciate the impact of our work. We address your concerns below:
> - *“it would be good to see results on more than one Atari environment”*
>     - Unlike traditional RL papers, we would need to construct a suitable true-proxy pair for each environment. For many games it is not obvious what reward to use other than game score, while other games run into technical difficulties because training hyperparameters from prior work were over-optimized for a particular model size. As a result, we focused on one game where we could overcome both difficulties.
> - *“The choice of proxy and true rewards for some of the environments seemed a bit arbitrary”*
>     - We address this in the common response.
> - *“I have the impression that the proposed benchmark requires a large amount of human feedback”*
>     - Our current benchmark actually does not require any human feedback. The benchmark we plan to release will consist of model checkpoints, their labels, and the trusted policy. In practice, one could use human feedback to create a trusted policy, but in our case we mark one of our trained policies as trusted and manually check that it does not exhibit reward hacking.
> If the benchmark were to be expanded to environments not considered in the paper, this would require humans to label which policies are anomalous and also label one of the trained policies to be a trusted policy. Labeling policies is very cheap (one only needs to check whether or not there is reward hacking), and would roughly cost: “1 minute / trajectory * 5 trajectories / policy = 5 minutes per policy”.
> - *“I also wonder whether using the distance from the (suboptimal) trusted policy would result in labeling a policy with superhuman performance on the true reward as aberrant”*
>     - Your example of AlphaGo’s Move 37 is very nice. We agree that using distance between policies may not lead to an optimal detector, especially when faced with aberrant actions. For this specific task, we envision a pipeline where such a move would indeed be flagged by the detector as an outlier but after some manual evaluation, humans do figure out the importance of this move and mark this as a “safe move”. With this, one can start exploring better policies which stay around these new safe policies.
>
> ### References
> [1] Christiano, P.F., Leike, J., Brown, T.B., Martic, M., Legg, S., & Amodei, D. (2017). Deep Reinforcement Learning from Human Preferences. NIPS.
>
> [2] Stiennon, N., Ouyang, L., Wu, J., Ziegler, D.M., Lowe, R.J., Voss, C., Radford, A., Amodei, D., & Christiano, P. (2020). Learning to summarize from human feedback. ArXiv, abs/2009.01325.

---

> > ### Comment · Reviewer_16uL · 2021-11-24
> > **Thank you for the response**
> >
> > Thank you for addressing my concerns, these are good points. I think I misunderstood the amount of human feedback required - it's good to hear that only labeling policies is needed.

---

### Official Review · Reviewer_uYeb · 2021-11-01

**Correctness:** 3
**Technical Novelty And Significance:** 3
**Empirical Novelty And Significance:** 2
**Recommendation:** 6
**Confidence:** 4

**Main Review:**

This paper targets the very important problem of misaligned AI models, under the specific lens of reinforcement learning based optimization. This problem has been widely studied in the AI safety / alignment community but has not received equivalent attention in the machine learning community. The experiments presented in the paper shed light on different settings where the problem of reward-hacking might happen, and under controlled variations of the agent's modeling power, and size of the state-space, shows different forms of reward-hacking in these environments. The paper is well-written and easy to follow, and has a thorough coverage of related works in AI alignment.

My main concerns with the paper are as follows:

1. None of the experiments in the paper are on sufficiently realistic settings. The paper tries to present empirical evidence of reward-hacking on a number of environments, but doesn't dig deeper into any of them. The specific challenges of each setting are different, and it would be useful to focus on a few settings and perform the evaluations keeping real-world considerations in mind. For example, in the traffic simulator, it would be useful to consider pixel-observations (which can be obtained easily for example in the CARLA simulator) and study the same problem.

2. The proxy reward functions considered are very "incorrect" in all the settings. In my understanding, the proxy reward functions are trying to show what happens when a reward function different from the true reward function is optimized by the agent. If so, then every attempt must be made to construct reward functions that are *very* close to the true reward function. This is similar to shaped reward functions in typical RL pipelines - it wis important to construct properly shaped functions. An example of this for the traffic example could be a weighted combination of the 4 proxy functions.

3. Missing experiments on environments where the true reward function cannot be easily constructed. The main challenge of misaligned models is in settings where the underlying reward function is unknown and cannot be queried. As a simple example, consider the task of pick and place of a puck with a robot arm. The end result of "puck in goal location" can be used to check task success but there is no underlying ground-truth dense reward function. In these settings, different papers come up with different shaped rewards for RL - it would be interesting to show experiments on these settings where the underlying reward function is not known, and demonstrate how different shaped rewards (similar to proxy rewards in the paper) fail, and draw inferences from them. This relates back to my first point above about realistic settings.

4. Question about atari experiments: why not evaluate on a large number of the atari games, as done by most RL papers that show results on atari?

5. Section 4 is very vague without clear takeaways. The notion of a trusted policy is introduced in section 4, as a way to mitigate the reward hacking problem, but the conclusions from Table 4 are unclear. What exactly is helpful in mitigating reward-hacking? Which metrics should we use for checking the extent of model mis-specification?

6. What is the number of training steps experiment measuring? Is the conclusion of this experiment that an agent that has been optimized for more time-steps will likely succumb to more reward-hacking? If so, then this requires addition experimentation because "not optimizing enough" cannot be a solution to reward-hacking, as that would also lead to lower true rewards as seen in the experiment.

7. The addition of action noise experiments (which the paper calls "resolution") seem to have a trivial takeaway? Adding more noise (i.e. higher variance) causes both the true and proxy rewards to decrease. I request the authors to explain how this non-trivial.

8. An additional dimension of agent capabilities to check would be the "type of training algorithm" used. This would likely be very important because different variants of algorithms (e.g. on-policy / off-policy, model-based / model-free) would have different failure modes with respect to reward hacking. This might provide more useful empirical takeaways for future work, compared to action resolution, and training steps variations considered.

In summary, I believe this paper tackles a very important and timely problem through a purely empirical lens. Since, this is a purely empirical paper, it is important to have more comprehensive experiments that are reflective of real-world settings, and have better motivated design choices. The paper in its current form does not provide useful takeaways that are practically significant and would require significant revision.



**Summary Of The Paper:**

This paper targets the very important problem of reward-hacking that occurs when the objectives optimized by intelligent agents are misaligned with respect to the tru objectives of the algorithm designer. The paper presents an empirical study across a range of different settings including a simple driving simulator, covid modeling, and a single atari game. The experiments show evidence of reward hacking as a function of modeling power of the agent and the size of the state-space. The paper concludes with some ideas and initial directions on how to potentially mitigate reward hacking.

**Summary Of The Review:**

I believe this paper tackles a very important and timely problem through a purely empirical lens. Since, this is a purely empirical paper, it is important to have more comprehensive experiments that are reflective of real-world settings, and have better motivated design choices. The paper in its current form does not provide useful takeaways that are practically significant and would require significant revision.

---

> ### Author Response · Authors · 2021-11-18
> **Response to Reviewer uYeb**
>
> Thanks for your detailed suggestions and feedback! We address your concerns below:
> - *“None of the experiments in the paper are on sufficiently realistic settings.”*
>    - We address this in the common response. In brief: most environments and reward functions are adopted from practitioners, and compared to the majority of past work our environments are significantly more complex.
> - *“In the traffic simulator, it would be useful to consider pixel-observations (which can be obtained easily for example in the CARLA simulator) and study the same problem.”*
>     - Using pixel observations would be an interesting environment to study. As mentioned in the common response, the traffic simulator we used in our work is widely used by practitioners. We believe this is evidence for its realism.
> - *“The proxy reward functions considered are very "incorrect" in all the settings.”*
>     - In the common response, we provide justifications of our choice of true and proxy rewards. We do not agree that the proxies are very “incorrect”, since in many cases they were adopted by practitioners. For instance, the average velocity proxy in the traffic simulator is the default reward in Wu et al., which has over 100 citations.
> - *“every attempt must be made to construct reward functions that are very close to the true reward function.”*
>     - In practice, reward designers do not often construct reward functions that are very close to the true reward function. For example, reward designers have tried to optimize YouTube’s recommender system for user engagement, using proxies such as clicks and watch-time. The difference between true and proxy reward functions in this instance is arguably more drastic than our traffic control example of mean velocity vs. mean commute time. Studies of YouTube have observed several unintended consequences including decreasing user happiness [1] and promoting conspiratorial behaviors [2]. We believe this shows that greatly misspecified reward functions do cause harm in practice and are worthy of study.
> - *“An example of this for the traffic example could be a weighted combination of the 4 proxy functions.”*
>     - We do consider a misweighting, which looks at a combination of the 4 proxy functions. This appears in Appendix A, Figure 7a and 7b.
> - *“Why not evaluate on a large number of the atari games, as done by most RL papers that show results on atari?”*
>     - Unlike traditional RL papers, we would need to construct a suitable true-proxy pair for each environment. For many games it is not obvious what reward to use other than game score, while other games run into technical difficulties because training hyperparameters from prior work were over-optimized for a particular model size. As a result, we focused on one game where we could overcome both difficulties.
> -*“the conclusions from Table 4 are unclear. What exactly is helpful in mitigating reward-hacking? Which metrics should we use for checking the extent of model mis-specification?”*
>     - In Section 4, we create an anomaly detection task using the trusted policy. We then propose three baseline anomaly detectors that measure the distance between the trusted policy and the potentially anomalous policy. Table 4 shows that none of the baselines perform well on all of the environments. We leave the task of designing a better anomaly detector for future work.
> - *“What is the number of training steps experiment measuring?”*
>     - Similarly to the other three settings, we vary training steps as one proxy for agent capabilities --- agents with limited resources can optimize the objective to a lesser amount than those with larger resources. Not optimizing enough is indeed not a solution to reward hacking, and that isn't what we were showing with these experiments. Instead, we propose that one solution would be to possibly regularize the optimized policy towards a safe policy, as indicated by our benchmarks.
> - *“The addition of action noise experiments (which the paper calls "resolution") seem to have a trivial takeaway?”*
>     - We agree that this result is unsurprising. As noted by reviewer bfGN, our goal was to avoid picking only the most convincing results. In the broader context of our work, these experiments were designed to measure one notion of agent capabilities.
> - *“An additional dimension of agent capabilities to check would be the "type of training algorithm" used.”*
>     - This is an interesting idea as a notion of agent capabilities. It may be tricky to implement in practice, because most training algorithms are not stable on most environments [3].

---

> > ### Author Response · Authors · 2021-11-18
> > **References for response to Reviewer uYeb**
> >
> > ### References
> > [1] Stray, J. Aligning AI Optimization to Community Well-Being. Int. Journal of Com. WB 3, 443–463 (2020). https://doi.org/10.1007/s42413-020-00086-3
> >
> > [2] Manoel Horta Ribeiro, Raphael Ottoni, Robert West, Virgílio A. F. Almeida, and Wagner Meira. 2020. Auditing radicalization pathways on YouTube. In Proceedings of the 2020 Conference on Fairness, Accountability, and Transparency (FAT* '20). Association for Computing Machinery, New York, NY, USA, 131–141. DOI:https://doi.org/10.1145/3351095.3372879
> >
> > [3] Henderson P, Islam R, Bachman P, Pineau J, Precup D, Meger D. Deep reinforcement learning that matters. In Proceedings of the AAAI conference on artificial intelligence 2018

---

> > > ### Comment · Reviewer_uYeb · 2021-11-20
> > > **Thanks for the response. I have updated my score.**
> > >
> > > Thanks a lot for the detailed response to reviewers' comments. My main concerns have been addressed, and I have revised the score. I would strongly encourage the authors to take into account all the reviewer suggestions for updating their paper, in particular those of reviewer bfGN.

---

### Official Review · Reviewer_bfGN · 2021-11-02

**Correctness:** 4
**Technical Novelty And Significance:** 3
**Empirical Novelty And Significance:** 3
**Recommendation:** 6
**Confidence:** 4

**Main Review:**

The paper’s findings are clearly useful to the research community as reward hacking has been a poorly understood but ubiquitous problem, broadly relevant for many practical settings. If the paper only brings extra attention to this topic, I think it has had some impact. The most important findings in my view are that:

1) Reward hacking can not only stop the true reward from increasing, but often actively lowers it
2) Increasing capability can increase reward hacking and thereby lower reward. This will be unsurprising to some readers but I believe it is still somewhat controversial and it helps to have demonstrations across several environments and several ways to increase capability, which makes these results more certain. This results in several interesting hacking behaviors.
3) Increasing capability can lead to phase transitions, a tricky new problem for that should be brought to the attention of RL practitioners as it may necessitate new monitoring strategies (such as the ones proposed in this paper).

Some readers won't find all of the above particularly surprising or novel. However, given how common and understudied reward hacking is, I believe that they deserve the more systematic study and exposition this paper provides.

The paper neglects to formalize some concepts. To some extent this is necessary for the first paper that investigates an important concept (reward hacking) which so far has no definition - I therefore think the lack of formality can be forgiven. What makes rigor more challenging here is that reward hacking and phase transitions can happen through multiple means such as increasing the size of the model and action space. I think the paper is a strong attempt at an important and hard problem, and will provide a good basis on which others can study reward hacking with increased formalization and rigor.

I agree with the other reviewers that some of environment and their proxy-true reward pair are not particularly convincing. I've lowered my score accordingly as I think this could detract from the impact this paper has on the research community. But I also recognize that it is difficult to design realistic pairs of reward functions. In reality, the true reward is typically too complex to specify (hence we need a proxy). Because a complex reward is not available in practice, in research papers we need to develop examples that are somewhat artificial.

While the paper is mostly well motivated, well written, and easy to follow, I do find that the presentation could be more precise and clear at times, as noted below.

In the final paragraph of the introduction, it was unclear to me what is the value proposition for the proposed benchmark and the baselines. This contribution could be better motivated.

I appreciate that in several places, such as Figure 4b,  the authors chose to display negative results where reward hacking and phase transitions were not observed. As researchers tend to cherry-pick the most convincing results, it is good to see that the authors avoided this.




———————————————— Detailed comments ————————————————



I’ll start by giving detailed feedback on the figures because most readers will look primarily at these.

- Figure 1:
—The figure with caption is not self-explanatory. E.g. who is the agent in this decision problem and what is their action space? (controlling a traffic light?)
—It’s not immediately obvious why the proxy and true rewards lead to different behavior here and it is not explained either. For an illustrative figure like this, I’d recommend using a more obvious example.
—The meaning of arrow colors is unclear
—It’s unclear what ‘optimization ability’ means here. If you’re referring to model size, you may want to use the same model size symbols you’re using in Fig 1c instead.

Figure 2: nice figure.

Figure 3: this figure is not self-explanatory, it needs more detail. The testing rate and the meaning of 16/112 are not explained. This figure uses a different format than figure 2 (separate plots for true/proxy rewards) which confused me at first sight. You could have one plot per model size to fix this, or move everything into one plot.

Figure 4: the takeaway could be clarified in the caption, especially for 4b (I believe it’s that no reward hacking occurred).
— Minor point: Figure 4a has a lower resolution than Figure 4b.

Figure 6 and Table 3 could also be improved to be more self-explanatory.


Minor suggestions:

“Overfitting their objectives” - unclear how overfitting is defined here, it seems to be a non-standard use of the word.

“we study how increasing optimization power affects reward hacking, by training RL agents with varying resources such as model size …” - increasing model size is not the same as increasing optimization. “Optimization power” makes me as the reader expect that you are talking about doing more/better gradient descent. Perhaps there is no better term here, in which case this is a necessary evil.

In Figure 4, it is unclear what action noise has to do with misspecification of the action space. Instead of adding Gaussian noise, I would expect that you vary the action space, e.g. discretize it at different resolutions. This does resemble Gaussian noise but it’s not the same thing.

The paper lacks motivation for why it’s hard not to resort to proxy rewards in practice. Some readers will already know this, others won’t.

COVID results (section 3.2): I get the result here and it is interesting - the policymaker only regulates once infections are high, but this saves little economic costs because it only delays the restrictions once while increasing the number of ICU patients indefinitely. But I found this not very well explained.

In section 3.2, it was unclear what is your takeaway from the Glucose environment. That reward hacking happens? That there was a phase transition?

“The objective of the RL agent is to promote a smooth traffic flow within the highway network” - this is confusing as in other places the true objective is said to be minimizing the mean commute time.


**Summary Of The Paper:**

This paper studies reward hacking, a common but understudied phenomenon, across a set of environments. Reward hacking emerges in several tasks, meaning that the resulting policy has a high proxy reward but a low true reward. A key finding is that reward hacking increases with agent capabilities so that increasing capability lowers the true reward. This holds across several ways of increasing capabilities (model size, training steps, action space, etc). The authors also find ‘phase transitions’ where a small increase in capability results in qualitatively new reward hacking behavior, a phenomenon that may require novel monitoring strategies. One such strategy is anomaly detection, for which the authors introduce a benchmark and baselines.


**Summary Of The Review:**

The paper’s findings are clearly useful to the research community as the reward hacking problem has been a poorly understood but ubiquitous. Three specific findings about reward hacking (see above) had to my knowledge not been empirically demonstrated in RL environments or only mentioned in passing but not studied explicitly. Before this paper, we therefore did not know if the findings correspond to generally applicable phenomena and through this paper we can gain some confidence. However, some environments and reward functions could be made more realistic.

---

> ### Author Response · Authors · 2021-11-18
> **Response to Reviewer bfGN**
>
> Thanks for the suggestions and for your positive comments on the importance of the topic! We address your concerns below:
> - *“The paper neglects to formalize some concepts”*
>     - Which concepts would you most like to see formalized? We will attempt to do so in our revision. We agree with the reviewer that our work is a first step and there are still other concepts whose precise definitions would still require a strong research effort from the community.
> - *“Some environments and reward functions could be made more realistic”*
>     - We address this in the common response. In brief: most environments and reward functions are adopted from practitioners and compared to the majority of past work our environments are significantly more complex.
> - *“In the final paragraph of the introduction, it was unclear to me what is the value proposition for the proposed benchmark and the baselines. This contribution could be better motivated.”*
>     - Our benchmark is motivated by our observations of phase transitions in the severity of reward hacking as agent capabilities increase. Phase transitions suggest that researchers may encounter sudden, unintended consequences during model deployment, which are important to avoid. The value of the benchmark is in allowing other researchers to make progress on reward hacking, particularly on detecting such phase transitions.
> - [Some Figures and Tables could be more self-explanatory]
>     - Thanks a lot for highlighting these issues. We have addressed the issues with Figures 1, 2, 3, 4, 6 and Table 3 in the updated draft.
> - *“increasing model size is not the same as increasing optimization”*
>     - We agree that these concepts are not similar. We previously used “agent capabilities” instead of “optimization power”, but it was pointed out that ‘capabilities’ appears more discrete rather than continuous. We would be happy to consider other terms to replace optimization power.
> - *“Instead of adding Gaussian noise, I would expect that you vary the action space, e.g. discretize it at different resolutions.”*
>     - Thank you for your suggestion; this is another approach for decreasing the resolution. We will update once we have experiments with these results.
> - *“The paper lacks motivation for why it’s hard not to resort to proxy rewards in practice.”*
>     - Thank you for the suggestion. We have updated the introduction with motivation on why designers often rely on proxy rewards.
> - *“In section 3.2, it was unclear what is your takeaway from the Glucose environment.”*
>     - We observed that reward hacking does occur but did not observe a phase transitions. We have updated Section 3.2 to more explicitly reflect the findings of Section 3.1.
> - *“this is confusing as in other places the true objective is said to be minimizing the mean commute time.”*
>     - Thanks; we have clarified this in the draft.
>
> We will reply again after updating the draft with the rest of your feedback.

---

> > ### Comment · Reviewer_bfGN · 2021-11-28
> > **Further minor comments**
> >
> > Thank you for addressing my comments.
> >
> > Some further minor comments to improve the final paper:
> >
> > > The value of the benchmark is in allowing other researchers to make progress on reward hacking, particularly on detecting such phase transitions.
> >
> > What I meant is that you could be more specific about how the benchmark provides value. Do you want other researchers to come up with methods that detect the (known) phase transitions in your benchmark without access to the true reward? Do you want other researchers to develop RL algorithms that don't have phase transitions? Overall, I'm not sure if the benchmark in its current form is adding as much value as the rest of the paper, and I could see it being more impactful if you expand it (as other reviewers noted) and made it into its own publication.
> >
> > > Which concepts would you most like to see formalized?
> >
> > Reward hacking and phase transitions. However, as I noted in my review this might be difficult at this early stage and I don't think this is strictly necessary.
> >
> > Typo: "proxies led misspecification". Further nitpick: optimizing a proxy does not lead to misspecification since the misspecification was already there before the proxy was optimized.

---

### Official Review · Reviewer_GVMn · 2021-11-03

**Correctness:** 4
**Technical Novelty And Significance:** 2
**Empirical Novelty And Significance:** 4
**Recommendation:** 6
**Confidence:** 4

**Main Review:**

The paper provides an interesting study of reward hacking behaviour where different parameters of the problem are varied to demonstrate the phenomenon. The authors chose a set of diverse and relevant environments. An interesting observation that might have large implications in practice is that training more capable agents might result in the behavior that achieves high proxy reward, but low true reward. While this behaviour was noticed before, I am not aware of a systematic study of the phenomenon where various parameters of the environment, reward and agent are controlled. Unfortunately, the paper does not provide any extensive related work overview that seems to be an important missing part given that the main contribution of the paper is the systematic study of reward hacking.

One concern that I have is that when the authors study misspecified rewards, the specification of the reward is only motivated by intuition. It is hard to appreciate how reasonable such a specification is without the full knowledge and experience with a particular environment. To me, a reasonable reward specification, even if it is misspecified, would still have significant correlation with the true reward. Sometimes (like in Figure 2) it seems that true reward goes down when the proxy reward goes up and it would be no surprise that a reward hacking occurs when the true objective is the opposite of what is optimised. A more interesting observation would be that a reward hacking occurs even when the objectives are mostly aligned and only sometimes diverge. To understand this better, it would be useful to look at some plots depicting the dependency of proxy and true reward on a broad set of sampled trajectories.

Finally, the authors provide a new baseline of anomaly detection for identifying the “phase transition” in the agent’s behaviour. Despite being small, such problem formulation could be useful for sparking more research in this direction.

Other comments and concerns:

- I didn’t find the Figure 1 very informative on its own, it is only possible to understand the bottom row after reading the main text and at that point the figure does not bring any new information. I would recommend trying to make the figure with its caption more self-contained

- Often rewards are manually crafted in continuous control tasks such as, for example, robotics. Would it be possible to provide another environment with, for example, control of a simulated robotic arm performing the manipulation tasks? The rewards might be assigned (and misspecified) based on the distances or positions of the objects.

- I would like the paper to be a bit more specific in its claims. For example, when the paper talks about more capable agents archiving “lower true reward”, lower than what? When the paper talks about “critical threshold”, it is the threshold of what?

- The results often seem to be quite noisy, for example, see Figure 2. How many experiments are conducted?

- The results would be more informative if they included the performance of the agent that optimises the true reward directly to provide an upper bound on the agent’s performance.

- I am not completely convinced by the example of diabetic risk and cost of insulin. I think this example does not take into account the ethics of such policy and the long-term costs of losing health due to acute hypoglycemic episodes.


**Summary Of The Paper:**

This paper provides a systematic study of “reward hacking” in the environments with the misspecified rewards. The authors conduct a set of experiments with 4 environments, several types of reward misspecification in each of them and several agents of different expressivity (model capacity). They notice that often the agents that are more capable end up obtaining high proxy reward, but low real reward. Besides, often the transition to the low real reward happens very quickly and authors call this phenomenon “phase transition”. Finally, they propose a baseline for anomaly detection to identify this phase transition.

**Summary Of The Review:**

I appreciate a systematic study of the reward hacking phenomenon where the parameters of the problem are manually varied. I think the related work overview should be extended to justify the “systematic” aspect of the paper. Besides, some information about the correlation between the true and the proxy rewards would be very beneficial to the reader without prior experience with the studied environments.

---

> ### Author Response · Authors · 2021-11-18
> **Response to Reviewer GVMn**
>
> Thanks for the suggestions and feedback on the paper! We address your concerns below.
> - *“the paper does not provide any extensive related work overview”*
>     - Thank you for the suggestion. We address this concern in the common response.
> - *“some information about the correlation between the true and the proxy rewards would be very beneficial to the reader without prior experience”*
>     - To clarify, are you asking for an experiment that quantifies the correlation between true and proxy rewards? We would be happy to compute this, although it would depend on the reference distribution chosen.
> - *“A more interesting observation would be that a reward hacking occurs even when the objectives are mostly aligned and only sometimes diverge.”*
>     - To clarify, are you asking for cases where the true and proxy reward are positively correlated for sufficiently small models, or just cases where the reward functions should intuitively be aligned? If the latter, we note that several proxy reward functions were adopted from practitioners (see common response for details): for instance, in the traffic environment “average velocity” is the reward used in the traffic simulator from Wu et al. (100+ citations), and we ourselves only noticed the problem with it when using large neural net policies.
> - *“I would like the paper to be a bit more specific in its claims.”*
>     - Thank you for those pointers; we have clarified the claims you mentioned in the abstract.
> - *“The results often seem to be quite noisy, for example, see Figure 2. How many experiments are conducted?”*
>     - We run our experiments with 3 seeds. We have updated Figure 2c after running with 5 seeds to reduce noise.
> - *The results would be more informative if they included the performance of the agent that optimises the true reward directly to provide an upper bound on the agent’s performance.”*
>     - We will add the performance of an agent trained on the proxy reward in the final draft.
> - *“I am not completely convinced by the example of diabetic risk and cost of insulin. I think this example does not take into account the ethics of such policy and the long-term costs of losing health due to acute hypoglycemic episodes.”*
>     - Yes, we agree that there are certainly ethical considerations to incorporate. Although the “true reward” has a morally positive connotation, we only adopt this terminology for convenience and are not making normative statements. Furthermore, there is prior work in the medical community [1-2] that studies income-related insulin underuse, where patients ration insulin (often at the cost to their own health) as a result.
>
> ### References
> [1] Herkert D, Vijayakumar P, Luo J, et al. Cost-Related Insulin Underuse Among Patients With Diabetes. JAMA Intern Med. 2019;179(1):112–114. doi:10.1001/jamainternmed.2018.5008
>
> [2] Fralick M, Kesselheim AS. The U.S. Insulin Crisis - Rationing a Lifesaving Medication Discovered in the 1920s. N Engl J Med. 2019 Nov 7;381(19):1793-1795. doi: 10.1056/NEJMp1909402. PMID: 31693804.

---

> > ### Comment · Reviewer_GVMn · 2021-11-22
> > **Thanks for your response and clarifications**
> >
> > I would like to thank the authors for their responses.
> >
> > *To clarify, are you asking for an experiment that quantifies the correlation between true and proxy rewards
> > To clarify, are you asking for cases where the true and proxy reward are positively correlated for sufficiently small models, or just cases where the reward functions should intuitively be aligned?*
> >
> > Yes, this is what I had in mind. I understand that it would depend on the state distribution, but it would still be informative to look at it with a few relevant state distributions, for example, states from a random agent, states from not very expressive agent or states from very expressive agent from this work. I understand that intuitively the rewards are supposed to mean the same, but I would like to see their positive correlation. I think it could also help to address the questions of the realism of the setting that other reviewers raised.
> >
> > Regarding the addition of the related work, I would prefer if some related work was included in the main paper as it would be much appreciated by any reader who is not an expert in this field.

---

> > > ### Author Response · Authors · 2021-11-23
> > > **Thank you for your clarification**
> > >
> > > Thanks for your clarification.
> > >
> > > *it would still be informative to look at it with a few relevant state distributions, for example, states from a random agent, states from not very expressive agent or states from very expressive agent from this work.*
> > >
> > > In an effort to showcase the correlation between the true and proxy rewards, we have added plots of the correlation between the proxy and true rewards in Appendix B. In short, we find that reward hacking does occur even when the rewards are positively correlated. We expand on this addition in our top-level comment.
> > >
> > > *I would prefer if some related work was included in the main paper*
> > >
> > > Due to space constraints, we have not updated the PDF to include more related work in the introduction. However, we agree with your suggestion that more related work should be included in the main paper and will update this in the final draft (there was not enough time to rework the introduction)

---

> > > > ### Comment · Reviewer_GVMn · 2021-11-23
> > > > **Thanks for the correlation plot**
> > > >
> > > > Thanks you for providing the correlation plots, I find them very curious. While it is of course expected that reward hacking would occur with negative reward correlation, I think it is an important observation that reward hacking occurs even with high correlation between the real and proxy rewards. I think adding such discussion to the introduction or conclusion (in the future) would strengthen the main point of the paper.

---

### Author Response · Authors · 2021-11-18
**Response to all reviewers**

Thanks to all the reviewers for their detailed and helpful feedback. We have updated the draft with a few changes (noted in the responses) and we reply when we update the draft with all requested changes.

Several reviewers have noted that Figure 1 with the caption is “not self-explanatory” and we have included a simpler version of it and improved the caption. We welcome any further feedback.

There were some concerns on how realistic our chosen environments and rewards are. To address those comments, we will:
- Detail prior works in reward hacking, which we will also add to the paper.
- Show that the design choices behind our environments and reward functions are motivated by real world applications and more complex than prior work.

## Related Work
Previous works in RL that demonstrate examples of reward hacking have often used grid-world type environments or game playing agents.
- Hadfield-Menell et al. [1] (NeurIPS oral) exhibit an example of reward hacking in a grid-world (Lavaland - 10x10) where the reward misspecification arises because of incorrect sensor reading at test time.
- Leike et al. [2] show two examples of reward gaming in grid-world environments. First, in a 3x3 boat race, the agent learns to move back and forth to collect points instead of finishing the race. Second, in a 5x7 tomato watering environment, the agent learns to put a bucket on its head so that plants appear watered.
- Toromanoff et al. [3] exhibit reward gaming examples in several Atari games (Elevator Action, Kangaroo, Bank Heist) where the agent keeps looping in a sub-optimal trajectory to obtain a repeated small reward.
- Baker et al. [4] study a multi-agent hide-and-seek environment. They show that, in the absence of a penalty for leaving the play area, the hiders learn to run endlessly to prevent the seekers catching them.
- Christiano et al. show reward hacking in the Pong game [5]. The agent learns to hit the ball back and forth instead of winning the point.
- Separate from grid-world or game playing applications, Stiennon et al. show reward hacking when learning to summarize Reddit posts [6]. An agent over-optimizes the learnt reward model and produces lower quality summarizations.

## Our Environments vs. Related Work
Our environments are more complex than the gridworld environments used in most related work, e.g. [1], [2], [3]. Our subjective judgment is that they are about as complex as the Atari environments in [4] and [5], although ours have more real-world grounding. For instance:
- The blood glucose environment relies on an FDA-approved simulator [7] used in medical research for testing glucose administration strategies.
- The traffic environment is based on a microscopic traffic simulator [12] (800+ citations) for urban planning. The simulator uses cars that are controlled by the Intelligent Driver Model, which is a widely-accepted approximation of human driving behavior [13] (3000+ citations).


## Realism of Proxies
Most of our proxies were either taken directly from practitioner code, or motivated by practical trade-offs. For instance:
- Our blood glucose proxy is taken from a previous work [8], which adapts the medical community’s measure of glycemic risk [9] into a reward function to train a continuous glucose controller.  Similarly, our true reward (economic cost of treatment) is motivated by studies [10-11] showing that low-income patients ration insulin to save money, i.e., patients may opt to emphasize monetary benefits over health benefits.
- For traffic, the “average velocity reward” is the reward used in the traffic simulator from Wu et al. (100+ citations), and we ourselves only noticed the problem with it when using large neural net policies.
- For COVID, different actors might emphasize political vs. medical costs. An economics paper [18] argues that “officials face incentives in making their pronouncements… overly pessimistic forecasts will grab media headlines and steer policy responses… [we] stress that the politics of economic policy cannot be ignored“. Similarly, a medical paper [19] uses ICU usage as an approximation of direct healthcare costs of the COVID-19 pandemic.

In particular, the traffic control, blood glucose, and COVID environments have misspecifications that may arise in practice (due to different stakeholders possessing misaligned incentives).

---

> ### Author Response · Authors · 2021-11-18
> **References for common response**
>
> ## References
> [1] Hadfield-Menell, D., Milli, S., Abbeel, P., Russell, S.J., & Dragan, A.D. (2017). Inverse Reward Design. NIPS.
>
> [2] Leike, J., Martic, M., Krakovna, V., Ortega, P.A., Everitt, T., Lefrancq, A., Orseau, L., & Legg, S. (2017). AI Safety Gridworlds. ArXiv, abs/1711.09883.
>
> [3] Toromanoff, M., Wirbel, É., & Moutarde, F. (2019). Is Deep Reinforcement Learning Really Superhuman on Atari? Leveling the playing field. arXiv: Artificial Intelligence.
>
> [4] Baker, B., Kanitscheider, I., Markov, T., Wu, Y., Powell, G., McGrew, B., & Mordatch, I. (2020). Emergent Tool Use From Multi-Agent Autocurricula. ArXiv, abs/1909.07528.
>
> [5] Christiano, P.F., Leike, J., Brown, T.B., Martic, M., Legg, S., & Amodei, D. (2017). Deep Reinforcement Learning from Human Preferences. NIPS.
>
> [6] Stiennon, N., Ouyang, L., Wu, J., Ziegler, D.M., Lowe, R.J., Voss, C., Radford, A., Amodei, D., & Christiano, P. (2020). Learning to summarize from human feedback. ArXiv, abs/2009.01325.
>
> [7] Man CD, Micheletto F, Lv D, Breton M, Kovatchev B, Cobelli C. The UVA/PADOVA Type 1 Diabetes Simulator: New Features. J Diabetes Sci Technol. 2014;8(1):26-34. doi:10.1177/1932296813514502
>
> [8] Magni L, Raimondo DM, Bossi L, Man CD, De Nicolao G, Kovatchev B, Cobelli C. Model predictive control of type 1 diabetes: an in silico trial. J Diabetes Sci Technol. 2007 Nov;1(6):804-12. doi: 10.1177/193229680700100603. PMID: 19885152; PMCID: PMC2769684.
>
> [9] BorIs. P. Kovatchev, Martin Straume, Daniel J. Cox & Leon.S Farhy (2000) Risk analysis of blood glucose data:Az quantitative approach to optimizing the control of insulin dependent diabetes, Journal of Theoretical Medicine, 3:1, 1-10, DOI: 10.1080/10273660008833060
>
> [10] Herkert D, Vijayakumar P, Luo J, et al. Cost-Related Insulin Underuse Among Patients With Diabetes. JAMA Intern Med. 2019;179(1):112–114. doi:10.1001/jamainternmed.2018.5008
>
> [11] Fralick M, Kesselheim AS. The U.S. Insulin Crisis - Rationing a Lifesaving Medication Discovered in the 1920s. N Engl J Med. 2019 Nov 7;381(19):1793-1795. doi: 10.1056/NEJMp1909402. PMID: 31693804.
>
> [12] P. A. Lopez et al., "Microscopic Traffic Simulation using SUMO," 2018 21st International Conference on Intelligent Transportation Systems (ITSC), 2018, pp. 2575-2582, doi: 10.1109/ITSC.2018.8569938.
>
> [13] Martin Treiber, Ansgar Hennecke, and Dirk Helbing. Congested traffic states in empirical observations and microscopic simulations. Physical review E, 62(2):1805, 2000.
>
> [14] "Flow: Architecture and Benchmarking for Reinforcement Learning in Traffic Control", C. Wu, A. Kreidieh, K. Parvate, E. Vinitsky, A. Bayen, arXiv preprint arXiv:1710.05465, 2017,
>
> [15] Buliung, R. N., & Kanaroglou, P. S. (2002). Commute minimization in the Greater Toronto Area: applying a modified excess commute. Journal of Transport Geography, 10(3), 177-186.
>
> [16] He, S., Peng, Y. & Sun, K. SEIR modeling of the COVID-19 and its dynamics. Nonlinear Dyn 101, 1667–1680 (2020). https://doi.org/10.1007/s11071-020-05743-y
>
> [17] Gary E. Weissman, Andrew Crane-Droesch, Corey Chivers, et al. Locally Informed Simulation to Predict Hospital Capacity Needs During the COVID-19 Pandemic. Ann Intern Med.2020;173:21-28. [Epub ahead of print 7 April 2020]. doi:10.7326/M20-1260
>
> [18] Boettke, P, Powell, B. The political economy of the COVID-19 pandemic. South Econ J. 2021; 87: 1090– 1106. https://doi.org/10.1002/soej.12488
>
> [19] Sarah M. Bartsch, Marie C. Ferguson, James A. McKinnell, Kelly J. O'Shea, Patrick T. Wedlock, Sheryl S. Siegmund, and Bruce Y. Lee The Potential Health Care Costs And Resource Use Associated With COVID-19 In The United States
> Health Affairs 2020 39:6, 927-935

---

> ### Comment · Reviewer_bfGN · 2021-11-21
> **Remaining concerns about reward functions**
>
> Thank you. My concern about the proxies used will be partially addressed by including the above motivations in the paper. However, I have not increased my score (6/10) because my previous evaluation already took into account that your proxies are motivated by real-world examples or prior work. (This may differ for other reviewers). I still feel that the paper could be improved by only using proxies that
> 1) a smart practitioner would actually use for training their agent
> 2) are more similar to the true reward or correlate with the true reward in most situations.
> Even better would be if
> 3) the true reward is actually difficult to specify and so we are forced to use a proxy. (As no true reward is given, reward hacking would have to be studied qualitatively.)
>
> (A further minor issue is that the paper is written for ML practitioners but some of the examples (Covid policy, glucose) the reward is likely "designed" by other people.)

---

> > ### Author Response · Authors · 2021-11-23
> > **Added correlation plots**
> >
> > Thanks for your feedback.
> >
> > In an effort to showcase the correlation between the true and proxy rewards, we have added plots of the correlation between the proxy and true rewards in Appendix B. In short, we find that reward hacking does occur even when the rewards are positively correlated. We expand on this addition in our top-level comment.
> >
> > We will look to add more examples of true-proxy reward misspecifications (especially the case when the true reward is difficult to specify), as this is a good suggestion.
> >
> > Thanks for your comment on the framing of the paper. We will add a section addressing this caveat during the introduction.

---

### Author Response · Authors · 2021-11-21
**Summary of Changes**

- Updated Figure 1 to be more self-contained and simpler
- Added Appendix A discussing related work, realism of environments, and realism of proxy rewards
- Added second paragraph in introduction on why proxies are often used in practice
- Removed Table 1 (There was some overlap between Table 1 and Table 2)
- Updated Section 3.2 COVID and Section 3.2 Glucose to be more clear and to more accurately reflect the findings of Section 3.1
- Changed occurrences of “smooth traffic flow” to “minimizing mean commute time”
- Made claims more specific in abstract
    - “Achieving lower true reward” → “Achieving lower true reward than less capable agents”
    - “critical thresholds” → “capability thresholds”
- Updated Figure 2c, Figure 7abc with more runs to reduce noise
- Updated Figures 1, 2, 3, 4, 6 and Table 3 for more clarity
- Added correlation plots in Appendix B

---

### Author Response · Authors · 2021-11-23
**Added correlation plots in Appendix B**

Reviewers bfGN and GVMn both requested correlation plots between the true and proxy rewards for different state distributions. We have added these in Appendix B. For now, only correlation plots for the traffic environment are present, but the full paper will have plot for all environments.

For a given model size, we obtain two checkpoints: one early in training (less than 1% of training complete) and one which achieves the highest proxy reward during training. We call the former the "random checkpoint" and the latter the "trained checkpoint". The random checkpoint is plotted in green and the trained checkpoint is plotted in blue.

For a given model checkpoint, we calculate the correlation $\rho$ between the proxy reward $P$ and true reward $T$ as the [Pearson correlation coefficient](https://docs.scipy.org/doc/scipy/reference/generated/scipy.stats.pearsonr.html) using 30 trajectories sampled from the model checkpoint. The states visited by one of the 30 trajectories are intended to be a rough estimate of the state distribution of the policy.

Interestingly, we see that reward hacking still occurs when there is positive correlation between the true and proxy rewards (bottleneck-misweight and merge-space misspecifications). Unsurprisingly, proxy-true pairs which are highly correlated (merge-misweight misspecification) do not exhibit reward hacking. Finally, proxy-true pairs which are negatively correlated (merge-ontological misspecification) exhibit the most reward hacking.

Thanks to the reviewers for this interesting suggestion. We have added these figures and explanation in Appendix B.

---

> ### Comment · Reviewer_bfGN · 2021-11-28
>
> Thank you for adding these. It could be interesting to visibly include correlation numbers or plots in the final main paper as the paper's impact depends on showing that the proxies and true rewards are sufficiently similar to be realistic.

---

### Decision · Program_Chairs · 2022-01-20

**Decision:**

Accept (Poster)

**Comment:**

I thank the authors for their submission and active participation in the discussions. All reviewers are unanimously leaning towards acceptance of this paper. Reviewers in particular liked that the paper is presenting an interesting and systematic study of reward hacking [GVMn] that is useful to the research community [bfGN] and targets an important problem [uYeb] in a rigorous way [16uL]. I thus recommend accepting the paper, but I strongly encourage the authors to further improve their paper based on the reviewer feedback, in particular in regards to improving positioning with respect to related work and a better formalization of their work.